# Risk of Absence of Measles Antibody in Healthcare Personnel and Efficacy of Booster Vaccination

**DOI:** 10.3390/vaccines9050501

**Published:** 2021-05-12

**Authors:** Chung-Jong Kim, Ji-Yun Bae, Kang-Il Jun, Hae-Sun Chung, Aeyeon Kim, Jihee Kim, Hee-Jung Son, Miae Lee, Hee-Jung Choi

**Affiliations:** 1Department of Internal Medicine, Ewha Womans University Seoul Hospital, Seoul 07804, Korea; jki342@gmail.com; 2Ewha Education and Research Center for Infection, Seoul 07985, Korea; jiyunbae@gmail.com (J.-Y.B.); sunny0521.chung@ewha.ac.kr (H.-S.C.); miae@ewha.ac.kr (M.L.); heechoi@ewha.ac.kr (H.-J.C.); 3Department of Internal Medicine, Ewha Womans University Mokdong Hospital, Seoul 07985, Korea; 4Department of Laboratory Medicine, Ewha Womans University Seoul Hospital, Seoul 07804, Korea; 5Office of Infection Control, Ewha Womans University Seoul Hospital, Seoul 07804, Korea; 41808s@eumc.ac.kr (A.K.); 41121s@eumc.ac.kr (J.K.); 6Office of Infection Control, Ewha Womans University Mokdong Hospital, Seoul 07985, Korea; 40757@eumc.ac.kr; 7Department of Laboratory Medicine, Ewha Womans University Mokdong Hospital, Seoul 07985, Korea

**Keywords:** measles, health personnel, vaccination

## Abstract

We aimed to identify the presence of the measles IgG antibody (mIgG-Ab) in healthcare personnel and finding out who needs the measles vaccination. The history of measles vaccination was obtained from the national vaccine registry. A baseline mIgG-Ab test was performed, and the measles vaccine was administered to participants who tested negative or equivocal for mIgG-Abs. During the study, 2885 (87.3%) of the 3303 employees were tested for measles serostatus. The baseline seropositivity rate for mIgG-Abs was 91.9%. Among the 234 seronegative cases, 82.9% were born after 1985. The seroprevalence rate was lower in those who received the measles–mumps–rubella (MMR) vaccine >10 years before the testing time, especially if they were born after 1985 and if there was only one previous record of vaccination. Among the 234 seronegative cases, MMR vaccination was administered in 174 cases, of which serostatus was evaluated in 146 cases. After the first dose, positive seroconversion was achieved in 126 participants (86.3%). After a second dose, 15 achieved (75.0%) positive seroconversion. In healthcare personnel born after the period when measles incidence significantly decreased, it may be necessary to reassess their immune status for measles if more than 10 years have elapsed since the last vaccination.

## 1. Introduction

Measles has re-emerged in recent times and its local epidemic outbreaks have frequently occurred worldwide [1,2,3,4]. Unlike measles in the past, the measles outbreaks that occurred recently in adults have developed in persons who had been properly vaccinated during their adolescence [5,6]. In South Korea, measles tends to recur mainly in children younger than 2 years of age and women in the age group 2–39 years [4,7]. This finding is supported by recently published data, which revealed a low rate of seropositivity for the measles antibody if the individuals tested were born after 1994 [8]. The low seropositivity rate in individuals born after 1994 persisted regardless of whether these individuals received two or more doses of the measles vaccine. Furthermore, it remains unclear as to whether the low seropositivity rate among these individuals is attributable to primary vaccine failure or waning of immunity over time.

Healthcare personnel are at a high risk of developing measles, owing to potential hospital exposure. In addition, the high reproduction ratio and shedding kinetics of the measles virus [9] increase the risk of large-scale in-hospital outbreaks when healthcare personnel develop measles [10]. Therefore, for the safety of patients and healthcare personnel to be ensured, it is necessary to determine whether healthcare workers have protective immunity against measles and whether those who do not have measles antibodies should be immunized again. 

Two doses of measles vaccination are recommended for healthcare personnel without presumptive evidence of measles immunity [11,12]. However, the appropriate number of immunizations in those who were previously immunized for measles remains unclear. Therefore, we evaluated the positivity rate of measles IgG antibody (mIgG-Ab) in healthcare personnel and risk factors of absence of measles immunity. We also evaluated that whether one or two vaccination doses are appropriate for adults without mIgG-Abs. In addition, we attempted to differentiate between primary vaccine failure and waning of immunity by using an avidity test for mIgG-Abs in those who developed immunity after vaccination.

## 2. Materials and Methods

### 2.1. Participants and Exclusion Criteria

This study was performed in all healthcare personnel at two university-affiliated hospitals in 2019. Both hospitals are located in Seoul, the capital city of the Republic of Korea, and are two university hospitals affiliated with one medical school. The two hospitals had 653 beds and 700 beds, respectively. The number of workers in each hospital was 1274 and 2029, respectively. Students were not included in the study, and the study was conducted only for employees who were employed by each hospital. The exclusion criteria for vaccination against measles were as follows: (1) pregnancy or future pregnancy planned within 2 months, (2) presence of an allergy to any antibiotic included in vaccine components, (3) history of a severe adverse event to a previous vaccination, and (4) lack of consent to participate in the study.

### 2.2. Study Protocol

The study was conducted in two stages. In the first stage, all employees were tested for baseline mIgG-Ab status. The baseline mIgG-Ab test was performed at the annual regular medical checkup or the medical checkup for new hospital employees. An enzyme-linked immunosorbent assay (ELISA; Liaison, DiaSorin, Italy) was used to determine the mIgG-Ab titer. The sensitivity and specificity of the kit were 97.4% (95% CI 94.1–99.2%) and 94.7% (95% CI 91.7–96.9%), respectively. The test results were interpreted according to the manufacturer’s protocol. Cut-off values were <13.5 AU (arbitrary unit)/mL (negative), 13.5–16.4 AU/mL (equivocal), and ≥16.5 AU/mL (positive). Furthermore, vaccination history was collected to determine previous immunization with the measles–mumps–rubella (MMR) or measles and rubella (MR) vaccine. Vaccination history was obtained from the national database on vaccine registration (available at https://nip.cdc.go.kr).

In the second stage, all employees negative for mIgG-Abs or who had equivocal results at baseline received the MMR vaccine except those who met the exclusion criteria. Those who agreed to participate in the study were evaluated to determine whether they had a positive seroconversion to mIgG-Ab 4 weeks after immunization (first follow-up). Personnel who did not agree to participate in the study received the second MMR dose 6–8 weeks after the first dose. Participants with positive seroconversion were considered to have completed the study and did not receive further vaccination. If mIgG-Ab test results were negative or equivocal at the first follow-up, a second dose of the MMR vaccine was administered. The mIgG-Ab test was repeated 4 weeks after the second vaccination (second follow-up). A measles IgG avidity assay (EUROIMMUN, Lübeck, Germany) was performed on the first samples from the vaccinated patients with positive seroconversion. The assay was conducted per the manufacturer’s protocol.

### 2.3. Subgroup Analysis

In South Korea, a large measles outbreak occurred between 2000 and 2001. Therefore, in 2001, a catch-up mass vaccination program was implemented among adolescents. Under this program, those born between March 1985 (age 35) and February 1994 (age 25) received a one-time catch-up immunization with the MR vaccine (Korea National Institute of Health 2001 guidelines for a measles mass vaccination campaign. Osong: KNIH, 2001). Since the reported catch-up vaccination rate in 2001 was higher than 97%, this group was likely vaccinated with the MR vaccine in addition to the routine MMR vaccination. As such, a sub-analysis was performed for this group.

### 2.4. Statistical Analysis

Baseline characteristics were compared between seropositive and seronegative employees. Additionally, baseline characteristics were compared between the initial MMR vaccine responders and non-responders, as a responder was defined as a participant who achieved seroconversion after the vaccination. All data were analyzed using SPSS software (ver. 22.0; SPSS Inc., Chicago, IL, USA). Continuous variables are expressed as mean ± standard deviation values. Student’s *t*-test or Mann–Whitney *U* test was used to analyze continuous variables. The χ^2^ or Fisher’s exact test was used to analyze categorical variables. All reported *p*-values were two-sided, and a *p*-value < 0.05 was considered statistically significant.

### 2.5. Ethics Statement

This study was approved by the institutional review board of the Ewha Womans University Mokdong Hospital and Seoul Hospital (IRB no. 2019-08-016).

## 3. Results

### 3.1. Baseline Characteristics and mIgG-Ab Seroprevalence

During the study period, 2885 (87.3%) of the 3303 employees were tested for measles serostatus. The baseline seropositivity rate for mIgG-Abs was 91.9% (2651/2885) (Figure 1). Employees who did not undergo the antibody test were significantly older than those who did (49.9 ± 12.3 vs. 34.9 ± 10.2 years, *p* < 0.001). Among the 234 seronegative cases, 82.9% (194/234) were born after 1985 (Appendix A). The baseline characteristics of personnel who tested positive and negative for mIgG-Abs are shown in Table 1. The seropositivity rates according to age and sex are presented in Figure 2 and Appendix A. The seropositivity rates according to occupation were as follows: doctors, 94.0% (425/452); nurses, 90.6% (1394/1538); other medical workers, 92.3% (369/400); and office workers or individuals in jobs without patient contact, 93.5% (463/495). The corresponding testing rates were 84.2% (452/537), 97.6% (1538/1576), 97.1% (400/412), and 64.6% (495/766), respectively.

### 3.2. History of Measles Vaccination

The seroprevalence rate of individuals positive for mIgG-Abs depended on whether a history of MMR vaccination existed. In participants born before 1985, the difference in the seroprevalence rates was not significant among known and unknown history of vaccination. However, in those born after 1985, the seroprevalence rate significantly differed depending on the number of previous vaccinations and the time elapsed since the last vaccination (Appendix A). The seroprevalence rate was lower when the last recorded MMR vaccine was administered more than 10 years ago, especially in those born after 1985. This was also true in those who received only a single vaccination dose in the past. However, the difference in the seroprevalence rate was not significant if there was a history of two or more previous vaccination doses (Table 2). 

### 3.3. Serologic Response to MMR Vaccination

Two hundred and thirty-four participants whose test results for mIgG-Abs were negative or equivocal were suitable for further study. Of the 234 participants, 138 had a history of measles vaccination. The measles vaccination history was unknown for the remaining 96 participants. Among the 234 subjects whose test results for mIgG-Abs were negative or equivocal at baseline, 146 were included in the MMR vaccination study. Eighty-eight patients were excluded or lost to follow-up (no response to the vaccine recommendation: 58; refusal for study participation: 28; pregnancy: 2). 

The first dose of the MMR vaccine was administered to 174 participants, and the antibody response was evaluated in 146 participants who agreed to participate in the study. Following the first MMR dose, 126 participants (86.3%) achieved positive seroconversion (responder) and 20 (13.7%) did not achieve seroconversion or had equivocal antibody results (non-responder). The demographic characteristics of those who had a positive or negative response to the first dose of the MMR vaccination are shown in Table 3. Among the 146 subjects whose antibody status was evaluated following the first vaccination, 7 (35.0%) of the non-responders and 71 (56.3%) of the responders had a history of MMR vaccination (*p* = 0.075).

A second MMR dose was administered to the 20 non-responders following the first vaccination. Five participants (25.0%) tested negative for mIgG-Abs following the second MMR dose and 15 (75.0%) tested positive. Three (60.0%) of the non-responders and four (26.7%) of the responders to the second MMR dose had a history of MMR vaccination (*p* = 0.176). Therefore, the overall seroconversion rate following MMR vaccination was 96.6% (141/146).

### 3.4. Subgroup Analysis: Population Subject to Catch-Up Vaccination in 2001

Of the 2855 employees who were tested for mIgG-Ab status, 1129 were born between March 1985 and February 1994. The baseline seropositive rate in this population was 91.1% (1029/1129). Among them, 759 had a recorded history of booster vaccination in 2001 and 370 did not have a vaccination record. The baseline seropositivity rates among those who did and did not receive a booster MMR vaccination were 92.4% (701/759) and 88.6% (328/370), respectively (*p* < 0.001). When those vaccinated after 2001 (not including those who underwent catch-up mass vaccination) were excluded, 919 of the 1129 participants remained. These participants had no recorded history of measles vaccination since 2001. In this population, the seropositivity rates among those who did and did not receive the booster in 2001 were 91.8% (593/646) and 86.8% (237/273), respectively (*p* = 0.020). 

In participants whose baseline mIgG-Ab test result was negative, the positive conversion rates following the first MMR dose were 86.7% (26/30) and 95.7% (22/23) in those who did and did not receive the 2001 booster, respectively. 

### 3.5. mIgG-Ab Avidity Assay

The antibody avidity assay was performed for 134 participants among 141 seroconverter, of whom 132 (98.5%) showed high avidity and two had equivocal results.

## 4. Discussion

In this study, we found that the baseline seropositivity rate of mIgG-Abs among healthcare personnel was 91.9%. We also found that 82.9% of the participants who tested negative for mIgG-Abs were born after 1985. The seronegativity rate was high in two population groups, especially in the younger age group who was born after 1985. The first group comprised participants whose MMR vaccination history was unknown, and the other group consisted of those whose last MMR vaccination had elapsed more than 10 years before study enrolment. Among those who received the MMR vaccine after negative or equivocal results, the mIgG-Ab seroconversion rates were 86.3% and 96.6% after the first vaccine dose and two vaccine doses, respectively. 

Since the introduction of MMR vaccination in South Korea in 1965, the annual incidence of measles has decreased in the country. Regardless, intermittent measles epidemics have still occurred in South Korea since 2000. The total number of measles infections between 2001 and 2019 was 24,363; with 23,060 (94.7%) cases reported in 2001 (https://www.cdc.go.kr/npt/biz/npp/ist/bass/bassDissStatsMain.do), which accounts for less than 0.04% of South Korea’s total population. Furthermore, most South Koreans achieved measles immunity through MMR vaccination, rather than measles infection [13]. Therefore, most of the population under the age of 35 years has acquired immunity through immunization. Considering the high rate of MMR and booster vaccinations in this age group during the year 2001, the reported seronegativity rate in this age group is astonishing.

The absence of measles antibodies in younger age group in general populations has been reported several times [8,14,15]. Kim et al. reported that the overall rate of individuals positive for the measles antibody was 73% among the healthcare personnel in a single tertiary hospital. However, in those who were born after 1994, it was 42%. Low seropositivity rates in those aged less than 30 years were also reported in other studies performed in South Korea [14,15]. However, the differences in seroprevalence according to age group vary by country. In a study conducted in the United States, the seropositivity rate for measles was lowest in the cohort of individuals born between 1967 and 1976. The rate gradually increased in older age groups, and individuals born between 1987 and 1997 had a positive seroprevalence rate of 93.3% [16]. In addition to country-specific variations, regional differences in seropositivity rates exist. In China, seropositivity rates for measles antibodies vary depending on the region. Pei et al. reported that more than 90% of individuals among subjects age over 8 months were seropositive in Shaanxi Province [17]. On the other hand, Boulton et al. reported that 10–20% of individuals aged 20–39 years in the Tianjin region were seronegative [18]. In a study conducted in the general population of South Korea, the seropositivity rates in those born between 1990 and 1994, and 1995 and 1998 were 69.6% and 48.5%, respectively [19]. The differences in the rates of measles seropositivity among different age groups and countries of origin may be attributable to differences in the strategy of MMR vaccination and measles prevalence. In South Korea, the live attenuated measles vaccine was introduced in 1965, and a free vaccination program was initiated in 1970. Thereafter, two vaccine doses were broadly used after the 1990s, and the catch-up vaccination program was conducted in 2001 [13]. As seen in other countries, it is estimated that even if the incidence of measles is low, a measles vaccination can secure a sufficient immunity to measles. Since measles immunity can be obtained by direct exposure to measles or measles vaccination, the low seropositivity rate for measles in people under age 35 years can be interpreted as a problem related to the low incidence of measles due to low exposure chance to wild type measles vaccine in the community and antibody maintenance after measles vaccination.

The results of our study suggest that birth after 1985 correlated strongly with higher rates of seronegativity for measles in South Korea. Furthermore, our study showed that vaccination history helped predict a negative mIgG-Ab test. Our study also suggests that if more than 10 years have passed since a person’s last MMR vaccination, the serostatus of mIgG-Abs should be re-evaluated in individuals born after 1985. The results of the avidity test suggested that the seronegative status of the study population was due to waning of immunity rather than primary vaccine failure. Almost all serum samples tested for mIgG-Ab avidity following positive seroconversion had a high avidity. The avidity test is used to differentiate primary and secondary vaccine failure in those previously vaccinated with MMR but develop measles infection [20]. Secondary vaccine failure is due to the waning of immunity over time. In previous studies, secondary vaccine failure contributed to the development of measles infection in previously vaccinated individuals [10,20]. Of note, the avidity test in this study was performed following vaccination and not measles infection. However, it was hypothesized that in a person with high avidity following positive seroconversion, the seronegative state before vaccination may be due to waning of immunity rather than primary vaccine failure.

Healthcare personnel in their 20s and 30s seem to be experiencing waning of immunity to measles infection, especially those whose vaccination status is unknown or whose last vaccination was more than 10 years ago [21]. Waning of measles immunity is more prominent in the case of vaccine-derived immunity than naturally obtained immunity, but the duration of waning immunity is still unclear [22]. One study reported that avidity of measles antibody was decreased by 8% when 20 years were elapsed after second MMR vaccination [23]. In South Korea, vaccination rates vary according to age group; however, the vaccination rate in the younger population is obviously higher than that in the older population. On the other hand, the incidence of measles infection in the general population was higher during the adolescence of the older age group than in the younger age group. Therefore, even if MMR vaccination is administered, measles immunity could wane considerably, resulting in seronegative conversion after more than 10 years among individuals born during measles elimination periods. Waning of measles immunity in the absence of boosting by the wild-type virus was also reported in South Korea [19]. For this reason, some researchers suggest that the third dose of MMR vaccination is needed in some settings such as outbreak [24]. Differences in seropositivity between males and females also support our hypothesis. In studies conducted in other countries, there was no difference in the measles seroprevalence according to sex [16,17,18,25,26,27,28,29,30,31,32,33]. However, in our study, m-IgG seropositive rate in males was higher than that of females. Because in South Korea all adult males over age 20 enlisted in the military have been vaccinated against MMR since 2012, and considering that the South Korean military is a conscription system in which most men are conscripted, it is highly likely that most men enlisted after 2012, who were born after 1992, received additional MMR vaccination. Therefore, for subjects born after 1994, the high seropositivity in males compared to females in the same age group might be the result of recent additional vaccination. Therefore, it may be necessary to reevaluate the measles immune status and consider MMR revaccination in these individuals.

This study had some limitations. First, we used the K-CDC national registration database to collect data on MMR vaccination history rather than personal memory. Although this database is widely used and all vaccinations included in the national vaccination program have been recorded, past data are often not well recorded. For example, the reported vaccination rate of the 2001 catch-up vaccination program was 97%; however, in our study, it was relatively lower (67.2%). Nevertheless, using a database to gather history improves the objectivity of the study compared to using memory-dependent methods. Second, in the 20 individuals who did not achieve mIgG-Ab seroconversion following the first MMR dose, it is necessary to consider the possibility that antibodies could develop as late as 6–8 weeks after vaccination. In a previous study, measles seroconversion occurred even later in some individuals, as late as after the administration of the third dose [34]. Therefore, some of the initial non-responders may have developed mIgG-Abs after the 4-week follow-up. This may have also occurred in the five participants who tested negative for mIgG-Abs following the second vaccination. Therefore, further long-term studies should be performed in these subjects to determine whether they ever develop mIgG-Abs. Third, in this study, the statistical power could not be improved due to the low proportion of subjects and non-responders who tested for antibodies after the first vaccination. However, since this study was performed for all employees belonging to two hospitals, there was a limit to increasing the total number of subjects for the purpose of increasing statistical power. In this perspective, those who were recommended for vaccination but did not receive the vaccination were an important issue. The measles vaccination rate in national immunization program for infant and child in the Republic of Korea was 98.2% [35]. However, in this study, 24.7% of those recommended for vaccination did not respond to recommendation. In a study of parents who refused measles vaccination in national immunization program, it seems that 54% refused vaccination because of distrust of the vaccine and 8% refused it because of concerns about adverse effects of the vaccine [36]. We think that the reason for refusal in our study is similar to this study, but additional research is needed on the reason for the relatively high non-response rate among adult healthcare personnel. Although it is not a legally mandated obligation to immunize medical personnel without measles immunity, efforts to increase vaccination acceptance will be needed in terms of the safety of patients and healthcare personnel.

## 5. Conclusions

In conclusion, in healthcare personnel born after the period when the incidence of measles decreased, it may be necessary to reassess the immune status to measles and consider MMR revaccination if more than 10 years have elapsed since the last dose.

## Figures and Tables

**Figure 1 vaccines-09-00501-f001:**
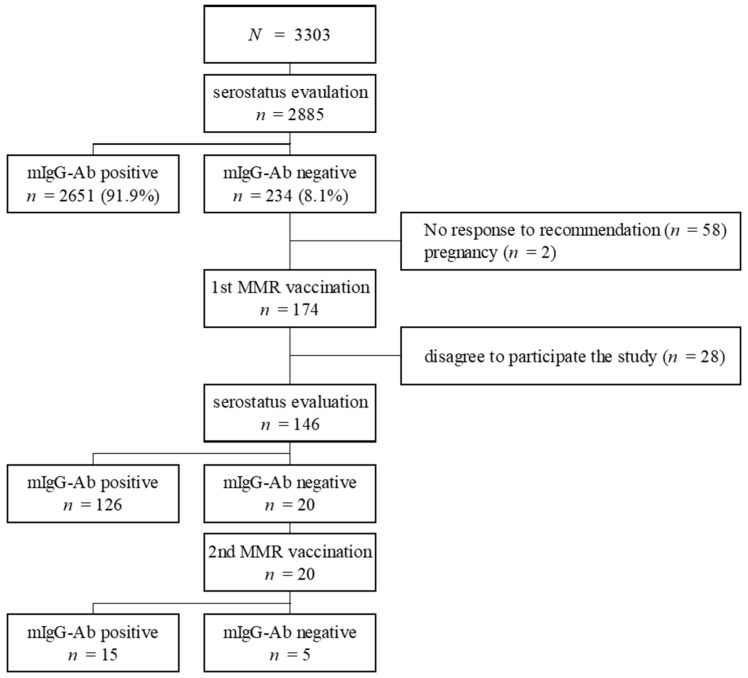
Flow chart of study. Note. MMR; measles–mumps–rubella vaccine, mIgG-Ab: measles immunoglobulin G antibody.

**Figure 2 vaccines-09-00501-f002:**
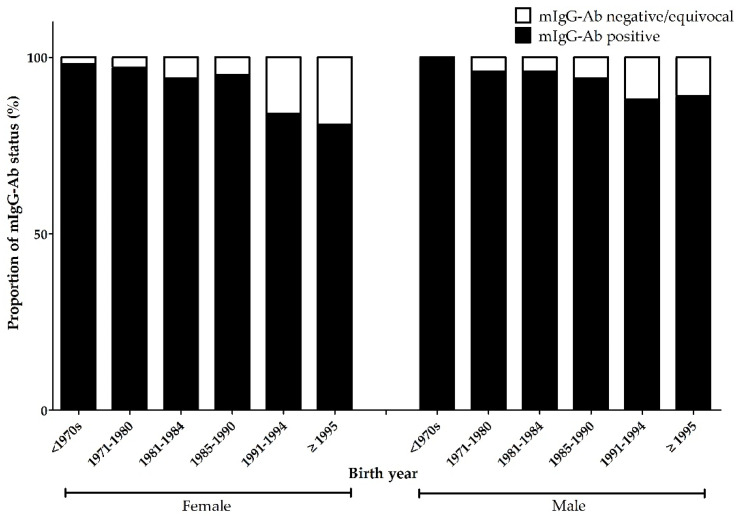
Proportion of measles IgG antibody-positive cases according to sex and birth year. Note. mIgG-Ab: measles immunoglobulin G antibody.

**Table 1 vaccines-09-00501-t001:** Baseline characteristics of participants according to measles IgG antibody status.

Variable	mIgG-Ab		*p*-Value
	Negative/Equivocal (*n* = 234)	Positive (*n* = 2651)	
Sex			
Male	32 (13.7%)	600 (22.6%)	0.001 *
Female	202 (86.3%)	2051 (77.4%)	
Age (years) (mean, SD)	28.1 (±6.6)	35.5 (±10.3)	<0.001 ^†^
Occupation			
Doctor	27 (11.5%)	425 (16.0%)	0.049 *
Nurse	144 (61.5%)	1394 (52.6%)	
Other medical personnel	31 (13.2%)	369 (13.9%)	
Non-patient contact	32 (13.7%)	463 (17.5%)	
Number of previously recorded vaccinations			
Birth prior to 1985			
Unknown	38 (95.0%)	1145 (93.0%)	0.849 *
Once	2 (5.0%)	81 (6.6%)	
Twice or more	0	5 (0.4%)	
Birth during or after 1985			
Unknown	60 (30.9%)	199 (14.0%)	<0.001 *
Once	122 (62.9%)	1033 (72.7%)	
Twice or more	12 (6.2%)	188 (13.2%)	

* χ^2^ test, ^†^ Student’s *t*-test. mIgG-Ab: measles immunoglobulin G antibody.

**Table 2 vaccines-09-00501-t002:** Seropositive rates of measles IgG antibody according to vaccination history.

Previous History of Vaccination		mIgG-Ab		Total	*p*-Value
Birth Year	Since Last Vaccination	Negative	Positive		
Unknown history of vaccination					
<1985		38	1145	1183	
≥1985		60	199	259	
Received single vaccination dose					
<1985	0–10 years	2 (100%)	80 (98.8%)	82	0.874 *
	Over 11 years	0	1 (1.2%)	1	
≥1985	0–10 years	6 (4.9%)	132 (12.8%)	138	0.011 ^†^
	Over 11 years	116 (95.1%)	901 (87.2%)	1017	
Received two or more vaccination doses					
<1985	0–10 years	0	5 (100%)	5	
	Over 11 years	0	0	0	
≥1985	0–10 years	8 (66.7%)	161 (85.6%)	169	0.078 *
	Over 11 years	4 (33.3%)	27 (14.4%)	31	

* Fisher’s exact test, ^†^ χ^2^ test. mIgG-Ab: measles immunoglobulin G antibody.

**Table 3 vaccines-09-00501-t003:** Comparisons between vaccine responders and non-responders following the first measles–mumps–rubella vaccination dose.

Variable	Negative after 1st Vaccination (*n* = 20)	Positive Conversion after 1st Vaccination (*n* = 126)	*p*-Value
Male	4 (20.0%)	13 (10.3%)	0.254 *
Birth year			
<1970s	0	0	0.728 ^†^
1971–1980	1 (25.0%)	2 (15.4%)	
1981–1984	0	0	
1985–1990	1 (25.0%)	4 (30.8%)	
1991–1994	1 (25.0%)	6 (46.2%)	
≥1995	1 (25.0%)	1 (7.7%)	
Female	16 (80.0%)	113 (89.7%)	0.254 *
Birth year			
<1970s	0	2 (1.8%)	0.692 ^†^
1971–1980	1 (6.3%)	7 (6.2%)	
1981–1984	1 (6.3%)	9 (8.0%)	
1985–1990	1 (6.3%)	11 (9.7%)	
1991–1994	5 (31.3%)	51 (45.1%)	
≥1995	8 (50.0%)	33 (29.2%)	
Previous vaccination history			
No history	13 (65.0%)	55 (43.7%)	0.156 ^†^
Received one dose	6 (30.0%)	67 (53.2%)	
Received ≥ 2 doses	1 (5.0%)	4 (3.2%)	
Length of time between last vaccination and current vaccination			
≤10 years	0	6 (8.5%)	1.000 *
≥11years	7 (100%)	65 (91.5%)	

* Fisher’s exact test; ^†^ χ^2^ test.

## Data Availability

Data available on request due to restrictions.

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
