# Peer review of "Risk of Absence of Measles Antibody in Healthcare Personnel and Efficacy of Booster Vaccination"

_vaccines, 2021, doi:10.3390/vaccines9050501_

Round 1

Reviewer 1 Report

The authors measure anti-measles antibody serological status in hospital employees evaluating multiple variables including age and prior immunization status. For seronegative or indeterminate individuals, they vaccinate up to two times and determine seroconversion rate. This study is very timely and relevant given the resurgence of measles and the likelihood of healthcare personnel to be exposed and to transmit infection. The study shows solid experimental design and reasonable interpretations, conclusions, and recommendations.

It would be helpful for the authors to clarify the comparisons yielding the stated P values in footnotes to the tables or in the text, in addition to the general description provided in the methods.

Author Response

Request 1

It would be helpful for the authors to clarify the comparisons yielding the stated P values in footnotes to the tables or in the text, in addition to the general description provided in the methods.

Response 1

In statistical analyses, we performed the Student's T-test for the continuous variable, and the χ2 or Fisher's exact test for the categorical variable. We have added the statistical methods to footnote in each table.

Reviewer 2 Report

Kim and colleagues assayed the presence of the measles IgG antibody (mIgG-Ab) in healthcare personnel to identify non-responders of measles vaccination in the healthcare settings. They used antigen testing to investigate the measles vaccination effectivity and to address the issues of waning immunity. They make recommendations of revisiting measles vaccination for non-responders.

I have a few minor suggestions:

  1. Since this test is antibody-based, the antibody should be assayed for its specificity by showing appropriate western blot gel.
  2. Population size and sample size for non-responder is very small to make these recommendations

Author Response

Reviewer 2.

Kim and colleagues assayed the presence of the measles IgG antibody (mIgG-Ab) in healthcare personnel to identify non-responders of measles vaccination in the healthcare settings. They used antigen testing to investigate the measles vaccination effectivity and to address the issues of waning immunity. They make recommendations of revisiting measles vaccination for non-responders.

I have a few minor suggestions:

Request 1

Since this test is antibody-based, the antibody should be assayed for its specificity by showing appropriate western blot gel.

Response 1

We used a commercial ELISA kit to test the presence of measles antibody. The measles antibody ELISA test used in this study was a liaison, Diasorin kit, and the sensitivity of the test presented by Diasorin, which supplies the kit, was 97.4% (95% CI 94.1-99.2%), and the specificity was 94.7% (95% CI 91.7- 96.9%). We have accepted the specificity of this kit and have added the study preferences in the manuscript. Currently, western blot for measles antibody is not performed in a commercial lab in Republic of Korea, and since blood samples were not stored separately in this study, it was not possible to confirm the specificity of the antibody through western blot.

From:

An enzyme-linked immunosorbent assay (ELISA; Liaison, DiaSorin, Italy) was used to determine the mIgG-Ab titer.

Page 2, Line 80-83

To:

An enzyme-linked immunosorbent assay (ELISA; Liaison, DiaSorin, Italy) was used to determine the mIgG-Ab titer. The sensitivity and specificity of the kit was 97.4% (95% CI 94.1-99.2%), and 94.7% (95% CI 91.7- 96.9%), respectively.

Request 2

Population size and sample size for non-responder is very small to make these recommendations

Response 2

The total number of subjects in this study was 2885, and a relatively large number (n=234) of them were sero-neagtive. However, among these, there were a few subjects who confirmed the seroconversion of antibodies after the first and second vaccination dose, so that a relatively small sample size was secured. In this study, it was impossible to further increase the number of study size due to we already enrolled all employees of both hospitals as the first analysis target. We think this is an important limitation of this study. Therefore, this point was explained in addition to the limitations of the study.

From: (-)

Page 10, line 329-335

To:

Third, in this study, the statistical power could not be improved due to the low proportion of subjects and non-responders who tested for antibody after the first vaccination. However, since this study was done for all employees belonging to two hospitals, there was a limit to increasing the total number of subjects for the purpose of increasing statistical power. In this perspective, those who were recommended for vaccination but did not receive the vaccination were an important issue. The measles vaccination rate in national immunization program for infant and child in republic of Korea was 98.2% [35]. However, in this study 24.7% of those recommended for vaccination did not respond to recommendation. In a study of parents who refused measles vaccination in national immunization program, it seems that 54% refused vaccination because of distrust of the vaccine and 8% refused because of concerns about adverse effects of the vaccine [36]. We think that the reason for refusal in our study is similar to this study, but additional research is needed on the reason for the relatively high non-response rate among adult healthcare personnel. Although it is not a legally mandated obligation to immunize medical personnel without measles immunity, efforts to increase vaccination acceptance will be needed in terms of the safety of patients and healthcare personnel.

Reviewer 3 Report

Estimated Authors,

Estimated Editors,

I've read with great interest the present paper from Kim et al. dealing with the risk of absence of measles antibody in healthcare personnel and efficacy of booster vaccination. Measles virus (MV) is a historically significant pathogen, that remains able to evolve in significant outbreaks around the world. As a consequence, vaccination is the mainstay of its management, and particularly in occupational settings.

Authors have performed an interesting research, that deals with this topic on two disparate sides. On the one hand, they have reported the immunization rate in a relatively large occupational study population of healthcare workers., assessing that a low but significant share of older HCW may be in fact not effectively immunized against MV. On the other hand, Kim et al have confirmed that a third dose may be effective in improving vaccination rates, with very low rates of non-responding subjects.

Despite its potential interest, some improvements are required, mostly formal ones.

Firstly, Authors should explain or at least discuss why Korean Females exhibit a more often unsatistying vaccination rate for MV. This is extensively confirmed by available data and by this research as well, but remains unexplained and not discussed across the text.

Second, some further details on the vaccination settings are required. What about the acceptance of MV in Korea? What about the critical issue represented by vaccine hesitancy in Korea and more precisely among Korean-HCWs?

Third: figure 2 is somewhat misleading. As the crude number of recruited HCWs of female gender is up to 4-5 times that of HCWs of male gender, and among the study participants the age groups are not equally represented, such figure should be redrawn through % values.

Fourth: all the tables are affected by some, often nothing more than annoying, formatting issue. Please double-check before the resubmission.

Fifth: some informations about the settings are forcibly required. At the moment, Authors have reported that: "This study was performed in all healthcare personnel at two university-affiliated hospitals in 2019". Unfortunately, no more informations are provided. Please report: the geographical location, specifying whether the aforementioned hospital are from metropolitan area or not, whether they are affiliated to universities or not, reporting whether some of the participants are in fact student of medical personnel involved in the formation courses, and eventually the size in terms of personnel and assisted patients of these hospitals.

Sixth: please clearly explain the occupational requirements for HCWs in Korea. I.e. there is any mandate for measles vaccine among HCWs? If a HCW without a certified MV refuses the vaccination, he/she can maintain his/her occupation in the healthcare settings or can be fired? This is particularly interesting, and the "laziness" of HCWs towards vaccinations contribute (in my opinion) to the often inappropriate vaccination rates.

Author Response

Reviewer 3.

Estimated Editors,

I've read with great interest the present paper from Kim et al. dealing with the risk of absence of measles antibody in healthcare personnel and efficacy of booster vaccination. Measles virus (MV) is a historically significant pathogen, that remains able to evolve in significant outbreaks around the world. As a consequence, vaccination is the mainstay of its management, and particularly in occupational settings.

Authors have performed an interesting research, that deals with this topic on two disparate sides. On the one hand, they have reported the immunization rate in a relatively large occupational study population of healthcare workers., assessing that a low but significant share of older HCW may be in fact not effectively immunized against MV. On the other hand, Kim et al have confirmed that a third dose may be effective in improving vaccination rates, with very low rates of non-responding subjects.

Despite its potential interest, some improvements are required, mostly formal ones.

Request 1

Firstly, Authors should explain or at least discuss why Korean Females exhibit a more often unsatistying vaccination rate for MV. This is extensively confirmed by available data and by this research as well, but remains unexplained and not discussed across the text.

Response 1

In this study, there was a difference in baseline measles antibody positivity according to sex. Existing studies referenced in this study, and other studies that were additionally searched, most of the cases did not show sex differences in data from other countries (as table below).

We though that the following reason might be influenced the seroprevalence. The Republic of Korea employs a conscript system, so most men are conscripted into the military service in their early twenties. In Korea, MMR vaccination has been given when enlisting in the military since 2012. Therefore, it is estimated that men born after early 1990s who enlisted in the military service after 2012 were more likely to receive the MMR booster vaccine within 10 years than women. Therefore, the difference in the positive rate of measles antibodies between men and women in Korea confirms once more that the recent measles vaccine booster has an effect on the antibody positivity rate. This was added to the discussion.

From:

Therefore, even if MMR vaccination is administered, measles immunity could wane con-siderably, resulted in seronegative conversion after more than 10 years among individuals born during measles elimination periods. Waning of measles immunity in the absence of boosting by the wild type virus was also reported in South Korea [19]. For this reason, some researchers suggest that 3rd dose of MMR vaccination is needed in some settings such as outbreak [24]. Therefore, it may be necessary to reevaluate the measles immune status and consider MMR revaccination in these individuals.

Page 10, line 299-314

To:

Therefore, even if MMR vaccination is administered, measles immunity could wane con-siderably, resulted in seronegative conversion after more than 10 years among individuals born during measles elimination periods. Waning of measles immunity in the absence of boosting by the wild type virus was also reported in South Korea [19]. For this reason, some researchers suggest that 3rd dose of MMR vaccination is needed in some settings such as outbreak [24]. Differences in seropositivity between male and female also support our hypothesis. In studies conducted in other countries, there was no difference in the measles seroprevalence according to sex [16-18, 25-33]. However, in our study, m-IgG se-ropositive rate in male was higher than that of female. Because in South Korea, all adult male over age 20 enlisted in the military have been vaccinated against MMR since 2012, and considering that the South Korean military is a conscription system in which most men are conscripted, it is highly likely that most men enlisted after 2012, who were born after 1992 received additional MMR vaccination. Therefore, for subjects born after 1994, the high seropositivity in male compared to female in the same age group might be the results of recent additional vaccination. Therefore, it may be necessary to reevaluate the measles immune status and consider MMR revaccination in these individuals.

Study title

Country

reference

male

female

p-value

Association between seroprevalence of measles and various social determinants in the year following a measles outbreak in Turkey

Turkey

Public Health. 2017 Jun;147:51-8.

81.5%

82.7%

0.590

Seroprevalence of measles, mumps, rubella, and varicella zoster virus antibodies among healthcare students: analysis of vaccine efficacy and cost-effectiveness

Turkey

Rev Esp Quimioter. 2019 Dec;32(6):525-31

82.8%

72.3%

0.097

Seroprevalence of measles antibodies and factors associated with susceptibility: a national survey in Mexico using a plaque reduction neutralization test

Mexico

Sci Rep. 2020 Oct 15;10(1):17488

99.2%

99.2%

0.412

Prevalence of Measles Antibodies in São José do Rio Preto, São Paulo, Brazil: A serological survey model

Brazil

Sci Rep. 2020 Mar 20;10(1):5179

84.1%

84.3%

0.975

Seroprevalence of Measles, Mumps, Rubella and Varicella Antibodies in the United States Population, 2009–2010

U.S.A

Open Forum Infect Dis. 2015 Jan;2(1):ofv006.

91.5%

92.4%

0.232

Low measles seropositivity rate among children and young adults: Asero-epidemiological study in southern China in 2008

China

Vaccine. 2010 Nov 29;28(51):8219-23

72.0%

69.1%

0.040

Seroprevalence of measles, mumps and rubella among young adults, after 20 years of universal 2-dose MMR vaccination in Israel

Israel

Hum Vaccin Immunother. 2015;11(6):1400-5

84.7%

86.9%

0.438

Measles and rubella seroprevalence in a population of young adult blood donors, France 2013

France

Epidemiol Infect. 2019 Jan;147:e109

90.5%

91.0%

0.780

Decreasing Seroprevalence of Measles Antibodies after Vaccination – Possible Gap in Measles Protection in Adults in the Czech Republic

Italy

PLoS One. 2017;12(1):e0170257

83.9%

82.8%

0.806

Identify the susceptibility profile to measles in the general population: Serological survey of measles antibodies in Shaanxi province, China, in 2016

China

Vaccine. 2017 Dec 19;35(52):7250-5

85.4%

86.6%

0.101

A population profile of measles susceptibility in Tianjin, China

China

Vaccine. 2016 Jun 8;34(27):3037-43

88.8%

87.4%

0.372

Request 2

Second, some further details on the vaccination settings are required. What about the acceptance of MV in Korea? What about the critical issue represented by vaccine hesitancy in Korea and more precisely among Korean-HCWs?

Response 2

Unfortunately, there are insufficient data on the acceptance rate of measles vaccine in healthcare personnel in Republic of Korea. In Korea, as in other countries, the measles vaccine is being vaccinated through national immunization program for infants and young children. In one study conducted in Republic of Korea, there was a study that investigated the reasons for refusal of parents to the vaccination in children. In the study, the reason of refusal of measles vaccination in their child was distrust of vaccination in 54%, worry of side effect in 8%, and belief that vaccination is unnecessary in 5%. There was no research that directly confirmed the reasons for refusal of measles vaccination targeting healthcare personnel, but we think the reasons will be similar.

As of 2019, the measles vaccine vaccination rate in infants and toddlers was 98.2% in cases where the second vaccination was completed, and it seems that the refusal of the measles vaccine in Republic of Korea is not high. However, 24.7% (58/234) of the subjects who were recommended for vaccination did not respond to the recommendation. Because they did not respond to our recommendation, we could not find out the cause of refusal of vaccination. Therefore, we think the reason for refusing vaccination is an important further research topic. We have added these in the discussion

From: (-)

Page 10, line 329-345

To:

Third, in this study, the statistical power could not be improved due to the low proportion of subjects and non-responders who tested for antibody after the first vaccination. However, since this study was done for all employees belonging to two hospitals, there was a limit to increasing the total number of subjects for the purpose of increasing statistical power. In this perspective, those who were recommended for vaccination but did not receive the vaccination were an important issue. The measles vaccination rate in national immunization program for infant and child in republic of Korea was 98.2% [35]. However, in this study 24.7% of those recommended for vaccination did not respond to recommendation. In a study of parents who refused measles vaccination in national immunization program, it seems that 54% refused vaccination because of distrust of the vaccine and 8% refused because of concerns about adverse effects of the vaccine [36]. We think that the reason for refusal in our study is similar to this study, but additional research is needed on the reason for the relatively high non-response rate among adult healthcare personnel. Although it is not a legally mandated obligation to immunize medical personnel without measles immunity, efforts to increase vaccination acceptance will be needed in terms of the safety of patients and healthcare personnel.

Request 3

Third: figure 2 is somewhat misleading. As the crude number of recruited HCWs of female gender is up to 4-5 times that of HCWs of male gender, and among the study participants the age groups are not equally represented, such figure should be redrawn through % values.

Response 3

As the reviewers pointed out, the current figure has limitations in delivering accurate information about the positivity of antibodies by age group. Therefore, we have changed the graph based on the percentage as pointed out.

Request 4

Fourth: all the tables are affected by some, often nothing more than annoying, formatting issue. Please double-check before the resubmission.

Response 4.

Before resubmission, the format of the table was checked and corrected.

Request 5

Fifth: some informations about the settings are forcibly required. At the moment, Authors have reported that: "This study was performed in all healthcare personnel at two university-affiliated hospitals in 2019". Unfortunately, no more informations are provided. Please report: the geographical location, specifying whether the aforementioned hospital are from metropolitan area or not, whether they are affiliated to universities or not, reporting whether some of the participants are in fact student of medical personnel involved in the formation courses, and eventually the size in terms of personnel and assisted patients of these hospitals.

Response 5

According to the points pointed out by the reviewer, detailed information on the two hospitals in which the study was conducted was presented.

From:

This study was performed in all healthcare personnel at two university-affiliated hospitals in 2019.

Page 2, line 67-72

To:

This study was performed in all healthcare personnel at two university-affiliated hospitals in 2019. Both hospitals are located in Seoul, the capital city of Republic of Korea, and are two university hospitals affiliated with one medical school. Each hospital has 653 beds and 700 beds. The number of workers in each hospital was 1,274 and 2,029. Students were not included in the study, and the study was conducted only for employees who were employed by each hospital.

Request 6

Sixth: please clearly explain the occupational requirements for HCWs in Korea. I.e. there is any mandate for measles vaccine among HCWs? If a HCW without a certified MV refuses the vaccination, he/she can maintain his/her occupation in the healthcare settings or can be fired? This is particularly interesting, and the "laziness" of HCWs towards vaccinations contribute (in my opinion) to the often inappropriate vaccination rates.

Response 6

There is no legislative regulation that can change the employment status of medical staff in relation to infectious diseases in the Republic of Korea. If the employee have an acute infectious disease such as active tuberculosis, the person must take a leave of work, but can return if his/her infectiousness is lost. Likewise, when entering a medical institution, there is no prohibition based on the state of immunity. However, many hospitals are checking the immune status of major infectious diseases, such as varicella, measles, tuberculosis, and hepatitis B, to protect staff and patients. In addition, if there is no immunity against these infectious diseases, vaccination is recommended and vaccination is provided at hospital expenses. Nevertheless, not getting vaccinated does not mean firing an employee.

Therefore, despite such support in the hospital, the management of unvaccinated employees and understanding the reasons why they were not vaccinated is an important issue in hospital infection control and staff management. This was added to the limitations of the study.

From:

(-)

Page 10, line 329-345

To:

Third, in this study, the statistical power could not be improved due to the low proportion of subjects and non-responders who tested for antibody after the first vaccination. However, since this study was done for all employees belonging to two hospitals, there was a limit to increasing the total number of subjects for the purpose of increasing statistical power. In this perspective, those who were recommended for vaccination but did not receive the vaccination were an important issue. The measles vaccination rate in national immunization program for infant and child in republic of Korea was 98.2% [35]. However, in this study 24.7% of those recommended for vaccination did not respond to recommendation. In a study of parents who refused measles vaccination in national immunization program, it seems that 54% refused vaccination because of distrust of the vaccine and 8% refused because of concerns about adverse effects of the vaccine [36]. We think that the reason for refusal in our study is similar to this study, but additional research is needed on the reason for the relatively high non-response rate among adult healthcare personnel. Although it is not a legally mandated obligation to immunize medical personnel without measles immunity, efforts to increase vaccination acceptance will be needed in terms of the safety of patients and healthcare personnel.

Round 2

Reviewer 3 Report

Estimated Authors,

Estimated Editors,

I congratulate with the Authors of this very interesting paper for having properly and extensively addressed all my concerns. I've no further requests, and therefore I warmly recommend the final acceptation of this paper.